# Suppressing dipolar relaxation in thin layers of dysprosium atoms

Pierre Barral [1,2] ✉, Michael Cantara[1,2], Li Du [1,2], William Lunden[1], Julius de Hond [1], Alan O. Jamison[1] & Wolfgang Ketterle [1]

The dipolar interaction can be attractive or repulsive, depending on the position and orientation of the dipoles. Constraining atoms to a plane with their magnetic moment aligned perpendicularly leads to a largely side-by-side repulsion and generates a dipolar barrier which prevents atoms from approaching each other. We show experimentally and theoretically how this can suppress dipolar relaxation, the dominant loss process in spin mixtures of highly magnetic atoms. Using dysprosium, we observe an order of magnitude reduction in the relaxation rate constant, and another factor of ten is within reach based on the models which we have validated with our experimental study. The loss suppression opens up many new possibilities for quantum simulations with spin mixtures of highly magnetic atoms.

Experiments with ultracold atoms or molecules are often limited by unfavorable inelastic collision rates. Several methods have been developed to control collisions such as isolating atoms in deep lattices[1], reducing collisional channels via confinement[2], or by mitigating their effects through the enhancement of elastic collisions via Feshbach resonances[3]. Polar molecules with electric or magnetic dipoles have been shielded from chemical reactions at short range by using repulsive interactions between dipoles, either in two dimensions or via microwave dressing[4–6].

Using dipolar shielding[7–9] with highly magnetic atoms is more challenging than with polar molecules as the dipolar interaction of the former is two orders of magnitude smaller than for the latter. Magnetic atoms have a simpler structure than molecules, allowing them to achieve lower temperatures while providing a controlled, tunable, and relatively simple platform for exploring novel forms of matter with long-range forces[10–16]. Dysprosium, with a magnetic moment of $10\mu_B$, has a magnetic dipole-dipole interaction that is 100 times larger than that of alkali atoms. However, dipolar relaxation – an inelastic spin-flip process that converts Zeeman energy into kinetic energy – occurs at a rate that scales as the square of the dipolar interaction, severely limiting the lifetime of any cloud with population in an excited Zeeman level. The dominance of dipolar relaxation has, thus far, precluded the experimental realization of many proposed new phenomena in spin mixtures of highly magnetic atoms[17–19]. Using dipolar shielding to prevent the atoms from undergoing dipolar relaxation requires a deep understanding of the dipolar interaction as it drives both the elastic and inelastic processes.

In this work we show that suppression of dipolar relaxation is possible since it occurs mainly at specific interatomic separations, where the dipolar potential reduces the wave function amplitude. It proves that confinement can not only affect the collisional channels between atoms[2], but also modify the interaction potential and provide shielding, as it was original proposed for molecules[7–9]. We observe an order of magnitude suppression of the dipolar relaxation rate, and, supported by comprehensive simulations of the decay rate[2], we show that another order of magnitude is within reach given reasonable parameters. In the limit of high magnetic fields, or for very low temperatures, the amount of suppression can be made arbitrary large. We first describe qualitatively the interplay of magnetic field, temperature, and shielding, then present our experimental results, followed by theoretical simulations.

## Results

### Basic principles of dipolar shielding

As mentioned, the dipole-dipole interaction is attractive in the case of a tip-to-tail orientation and repulsive for the side-by-side one. Constraining atoms to an $xy$ plane, with a magnetic moment aligned perpendicularly along $z$, leads to a largely side-by-side repulsion and generates a dipolar barrier. The dipolar length $a_{dd} = \frac{\mu_0}{4\pi} \frac{\mu (10\mu_B)^2}{\hbar^2}$ represents the strength of the interaction and the two-particle oscillator

---

[1]Research Laboratory of Electronics, MIT-Harvard Center for Ultracold Atoms, and Department of Physics, Massachusetts Institute of Technology, Cambridge, MA 02139, USA. [2]These authors contributed equally: Pierre Barral, Michael Cantara, Li Du. ✉e-mail: pbarral@mit.edu

length $a_z = \sqrt{\hbar/\mu\omega_z}$ the extension of the cloud in the $z$ direction. We denoted $\mu$ and $\omega_z$ the reduced mass and the trap frequency respectively. A dipolar barrier appears when the dipolar length $a_{dd} > 0.34 a_z$[9], which we refer to as the quasi-2D regime. Thus, experiments with dysprosium require 10,000 times higher axial frequencies than polar molecules to compensate for the 100 times smaller dipolar length. Our experiments have reached this regime with $a_z = 20$ nm and $a_{dd} = 10$ nm.

Three parameters determine the loss rate in quasi-2D: the ratio $a_{dd}/a_z$ set by the confinement, the temperature $T$, and the magnetic field $B$. The potential barrier increases with confinement, ultimately reaching the pure-2D limit as $a_z \to 0$ as shown Fig. 1a. As the temperature decreases, the wave function of an incoming pair is suppressed by

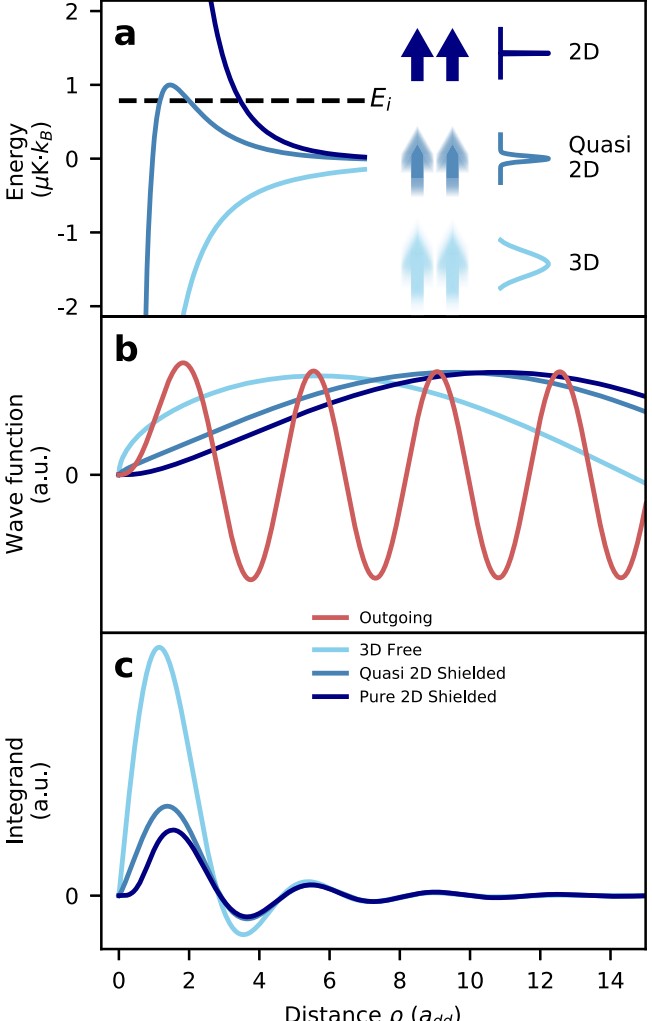

**Fig. 1 | Principle of dipolar shielding. a** Effective radial potential between two atoms from equation (6) for: no confinement (light blue, 3D), quasi-2D with $\omega_z/2\pi = 300$ kHz (steel blue) and pure-2D (dark blue). The incoming energy is given by the temperature $T = 1\,\mu K$. **b** Wave function solutions of equation (6) with $n = 0$ and initial orbital momentum $m_i = 0$ for the three confinement strengths described above, and in red the spin-flipped outgoing wave function to $m_f = 2$ and $B = 500$ mG. The effect of shielding of the outgoing wave function is negligible for these parameters. **c** Integrand of Fermi's golden rule (equation (2) and see equation (SI-10) in Supplementary Information Note 2). Each curve is the product of the respective wave function in **b**, the outgoing wave function, and the double spin-flip operator from equation (4) integrated with the harmonic oscillator wave functions in the $z$-direction. The shielding we implement here corresponds to the difference between the light blue and steel blue curves. The minimum attainable decay rate for this incoming energy corresponds to the dark blue curve. See Supplementary Information Note 5 for insights on the behavior of the integrand.

the barrier over a longer range, thereby decreasing the chance of two atoms reaching close range. This shielding effect on the wave function is illustrated in Fig. 1b. As the magnetic field increases, the range where dipolar relaxation occurs is shortened and the shielding increases. Indeed, a higher magnetic field leads to a higher released energy, and correspondingly a more rapidly oscillating outgoing wave function (see red curve Fig. 1b). Since the dipolar potential falls off as $1/r^3$, the majority of the decay will come from the first oscillating lobe of the outgoing wave function, as seen in Fig. 1c. The range of dipolar relaxation, therefore, decreases as the magnetic field increases. This can also be explained in a semi-classical picture: the Franck-Condon principle predicts spin flips to occur at the classical turning point of the outgoing wave function[2,20], i.e. when the released Zeeman energy equals the energy of the centrifugal barrier. Correspondingly, higher magnetic fields cause spin-flips to occur at a shorter range, ultimately behind the barrier felt by the incoming atoms, where they are strongly suppressed. In addition to increasing shielding, magnetic fields affect dipolar relaxation rates via the density of final states. Without shielding in 3D and 2D[2,19,21], this leads to an increasing rate (for bosons). Shielding qualitatively changes the magnetic field dependence of the dipolar relaxation rate which now decreases with magnetic field (see Supplementary Information Note 4).

## Experiment

Here we study these principles experimentally. We load ~$8 \times 10^4$ spin-polarized $^{162}$Dy atoms in the excited $|J = 8, m_J = 8\rangle$ Zeeman level (see Methods for details) in an optical lattice and get a stack of about 45 thin pancakes ('crêpes'). The crêpes reach an $a_z/2 = 10$ nm root-mean-square (RMS) width and a 5.7 $\mu$m radius. The peak density is $2.9 \times 10^9$ cm$^{-2}$. The experiment is performed at $T \approx 1.6\,\mu K$, above the BEC transition temperature (300 nK), to prevent convolving our results with changes in the two-particle correlation function[22,23]. The quantization axis is set by an external magnetic field along the $z$ direction. The lattice beam is blue detuned, with its radial repulsion compensated by a coaxial red-detuned optical dipole trap, as shown in Fig. 2a. Axial trap frequencies are limited to $\omega_z/2\pi = 260$ kHz by the maximum laser power of the compensation beam.

By measuring the atom losses we determine the inelastic decay coefficient, $\beta_{3D}$, as defined by the differential equation for the 3D density $n$:

$$\frac{dn}{dt} = -\beta_{3D} n^2. \tag{1}$$

We obtain densities from the measured atom number, temperature and trap frequencies, and average over the stack of crêpes (also see the Methods section). We sometimes refer to the 2D loss rate $\beta_{2D}$ in cm$^2$/s, which uses the 2D density instead. It is related to $\beta_{3D}$ through the axial harmonic confinement via $\beta_{2D} = \beta_{3D}/(a_z\sqrt{\pi})$.

Our experimental results are shown in Fig. 2b, c. We also compare the theoretical shielded decay rate (solid blue) with the one we would expect in the same crêpe geometry if there was no elastic dipolar potential to repel the atoms (dashed blue). In contrast to the loss rate in a 3D geometry (red), which increases with $\sqrt{B}$ (see Supplementary Information Note 5), we observe the signature of shielding in Fig. 2b: a much weaker dependence on magnetic fields (solid blue). Our results are also qualitatively different from those presented in[2], represented by the dash-blue curve. Their approach relies solely on shaping the trap to modify the available outgoing channels, whereas we go further by altering the interaction potential experienced by the atoms.

We operate in the quasi-2D regime which differs from the pure-2D one in several aspects. Compared to pure-2D, the finite axial extent of the quasi-2D geometry softens the radial barrier, reducing the barrier height to energies comparable to typical temperatures in the experiment. Furthermore, for Zeeman energies that are larger than the axial

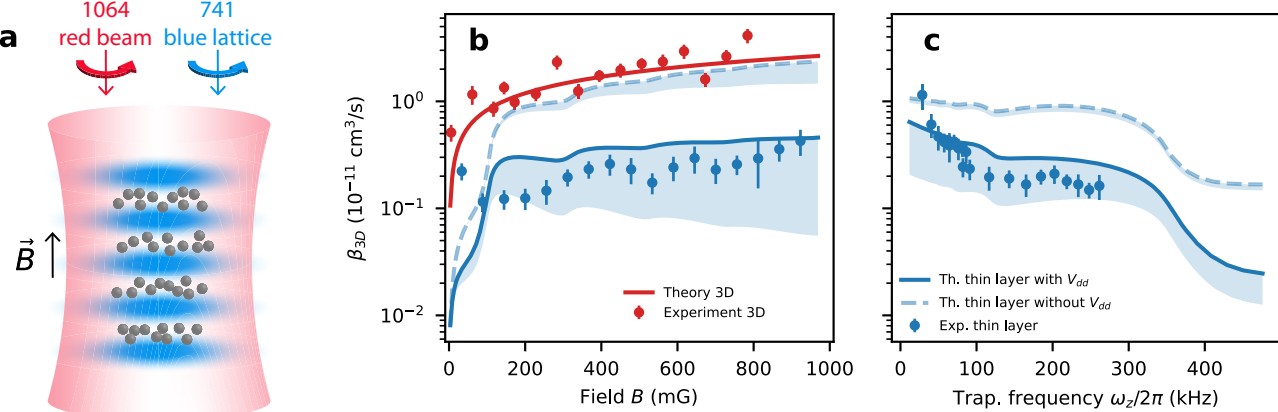

**Fig. 2 | Experiment scheme and results. a** Trap geometry. A blue-detuned 741 nm retroreflected beam repels the atoms to create a 1D lattice. The finite contrast of the lattice and the zero-point motion of the atoms in the ground state create a repulsive transverse potential, which is compensated by a 1064 nm red-detuned beam to create an adjustable transverse harmonic confinement. **b**, **c** Experimentally measured $\beta_{3D}$ in a large volume trap (red) and in a thin layer (blue). The lines are theory curves obtained by using Fermi's golden rule (see Supplementary Information Note 2 for derivations). The red curve shows the decay rate in 3D[19,21], the dashed blue curve is for non-shielded atoms in a lattice[2]. The solid blue line takes into account the shielding induced by the elastic dipole-dipole interaction. All theoretical curves are thermally averaged over the incoming momenta. The shaded blue region corresponds to the inclusion of contact interactions (see an extended discussion of the short-range physics in Methods). **b** Measurement of $\beta_{3D}$ as a function of magnetic field. The axial trap frequency is $\omega_z/2\pi = 185$ kHz which corresponds to $a_z/2 = 13$ nm. **c** Measurement of $\beta_{3D}$ in a constant magnetic field of 200 mG while varying the trap frequency. The uncertainties are set by the atom number stability, cloud temperature measurement and trap frequency measurements (see Methods section). The relaxation rates measured at very low fields deviate from the theoretical values, most likely because the imperfect circular polarization of the lattice and compensating beams changes the orientation of the dipoles (see Methods section).

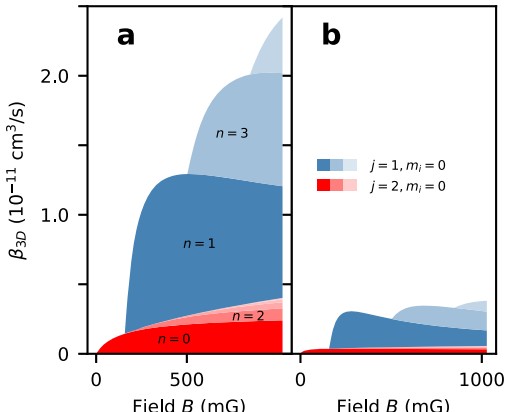

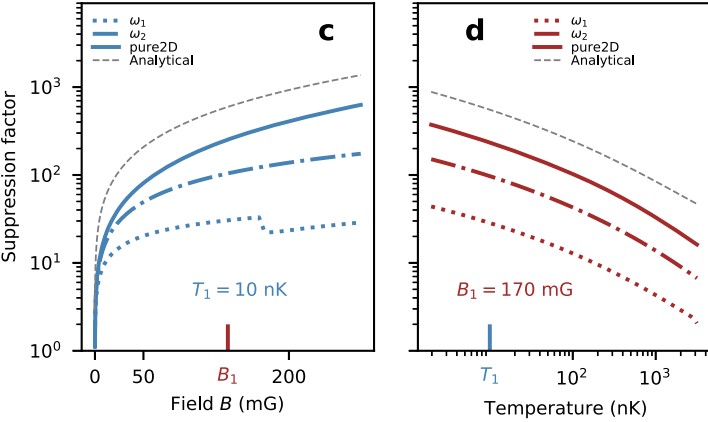

**Fig. 3 | Theoretical loss rate coefficients.** Channel-by-channel decomposition of the dipolar relaxation rates of the $m_i = 0$ incoming state (valid when $\mu_B B \gg k_B T$, see Supplementary Information Note 3) in a 300 kHz trap, both for free wave functions (**a**) and shielded ones (**b**). The blue and red colors correspond to single and double spin flips, respectively. The different shades correspond to different harmonic oscillator states as they open up with increasing magnetic field. The suppression factor defined as the ratio of $\beta_{3D}$ obtained from shielded and free wave functions, both at fixed temperature $T_1$ (**c**), and at fixed magnetic field $B_1$ (**d**). For each of the graphs we present curves for $\omega_1/2\pi = 300$ kHz (dotted), $\omega_2/2\pi = 1.8$ MHz (dashed dotted), the pure-2D case (solid line) as well as the analytical approximation (grey dotted) from equations (SI-17) and (SI-22) detailed in Supplementary Information Note 4.

trapping frequency, new collisional channels open, with a portion of the released energy converted into axial excitation and the remainder into radial motion. As a result, the relaxation for these processes is shifted to larger distances, thereby weakening the shielding. The first channel opening is visible in Fig. 2b around 100 mG as well as in Fig. 3a, b. The aforementioned factors lead to a relaxation rate that does not decrease with magnetic field, as it would in the pure-2D case, but instead shows a weaker increase compared to the case without a dipolar barrier (dashed blue). Figure 2c shows the loss rate coefficient as a function of axial confinement. The loss rate decreases with confinement due to enhanced shielding by the dipolar repulsion and closing axially excited states channels.

We have reduced the loss rate coefficient to approximately $1 \times 10^{-12}$ cm³/s. Over a large range of magnetic fields in a lattice, we

achieved more than an order of magnitude reduction in the dipolar relaxation rate coefficient compared to the unshielded case. The agreement between the numerical calculations and the experiment enables extrapolation beyond the current limitation of the experiment: very favorable loss rate coefficients of $2 \times 10^{-13}$ cm³/s can be achieved at 200 mG with an axial confinement of 500 kHz at 1 μK. This matches the lowest rate obtained with fermions through Pauli suppression in reference[19]. Under such conditions, axial excitations are energetically forbidden and the 2D decay rate is less than a factor of 3 above the pure-2D limit. By lowering the temperature to 100 nK, the relaxation rate would be suppressed by an additional factor of three and reach the $10^{-14}$ cm³/s regime. To further understand how these numbers are computed, we describe our theoretical model in the following paragraphs.

### Theoretical model

Dipolar relaxation rates can be calculated from Fermi's golden rule. The decay rate Γ of 2 particles is given by

$$\hbar\Gamma = 2\pi |\langle \Psi_{\text{out}} | \hat{V}_{\text{dd}} | \Psi_{\text{in}} \rangle|^2 \rho(E), \qquad (2)$$

where $\rho(E)$ is the final density of states at energy $E$. The incoming wave function is an excited Zeeman state with transverse momentum $\boldsymbol{k_i}$ in the lowest harmonic oscillator state, $n_i = 0$. The outgoing wave function is a lower Zeeman state with momentum $\boldsymbol{k_f}$ in the harmonic oscillator state $n_f$. The loss rate coefficient $\beta_{2D}$ is related to Γ through $\beta_{2D} = \pi L^2 \Gamma$, with $L$ being the radius of the transverse box used to normalize the wave functions. The atoms are coupled by the magnetic dipole-dipole interaction:

$$\hat{V}_{\text{dd}} = \frac{\mu_0}{4\pi}(g_J\mu_B)^2 \frac{\hat{\boldsymbol{J_1}} \cdot \hat{\boldsymbol{J_2}} - 3(\hat{\boldsymbol{J_1}} \cdot \boldsymbol{u_r})(\hat{\boldsymbol{J_2}} \cdot \boldsymbol{u_r})}{r^3}, \qquad (3)$$

where $\boldsymbol{r}$ is the interatomic separation (with corresponding unit vector $\boldsymbol{u_r}$). The magnetic field points along $z$. Atoms in the initial spin state $|j_0\rangle = |m_{J_1} = 8, m_{J_2} = 8\rangle$ can collide and remain in the same spin state, or relax to either $|j_1\rangle = (|7,8\rangle + |8,7\rangle)/\sqrt{2}$ or $|j_2\rangle = |7,7\rangle$. The dipole-dipole operator acting on $|j_0\rangle$ is:

$$\hat{V}_{\text{dd}}|j_0\rangle = \frac{\mu_0(Jg_J\mu_B)^2}{4\pi r^3}\left[(1 - 3\bar{z}^2)|j_0\rangle - \frac{3\bar{z}\bar{r}_+}{J^{1/2}}|j_1\rangle - \frac{3\bar{r}_+^2}{2J}|j_2\rangle\right] \qquad (4)$$

$$= V_{\text{dd},0}|j_0\rangle + V_{\text{dd},1}|j_1\rangle + V_{\text{dd},2}|j_2\rangle \qquad (5)$$

with $\bar{z} = z/r$ and $\bar{r}_+ = (x + iy)/r$. Equation (4) shows the three effects of the dipolar interaction: an elastic scattering process, a single spin-flip proportional to $\bar{z}$, and a double spin-flip which implicitly depends on $z$ through $r$.

In the two-dimensional limit where $z = 0$, the single spin-flip term vanishes and the elastic term is a purely repulsive $1/\rho^3$ potential (where $\rho = \sqrt{x^2 + y^2}$). This potential has an analytic solution at zero temperature ($k_i = 0$)[24], while other cases have to be solved numerically.

We assume a quasi-2D geometry where we ignore the effect of $V_{\text{dd},0}$ on the $z$ motion, which is then factorized and described by harmonic oscillator wave functions (see Methods for a discussion on this approximation). The elastic portion of the operator in equation (4) is averaged over the $z$ direction. This leads to an effective repulsive potential (see Fig. 1a) in the one-dimensional radial Schrödinger equation:

$$\left\{\frac{\hbar^2}{2\mu}\left(-\frac{d^2}{d\rho^2} + \frac{m^2 - 1/4}{\rho^2}\right) + \langle n|V_{\text{dd},0}|n\rangle\right\}\phi = \frac{\hbar^2 k_i^2}{2\mu}\phi. \qquad (6)$$

Here, the state $|n\rangle$ is the $n^{\text{th}}$ harmonic oscillator's state along $z$. We focus on incoming states with zero projection of orbital angular momentum, $m_i = 0$, as this channel dominates for any reasonable magnetic field (see Supplementary Information Note 3).

We solve the Schrödinger equation for the radial wave function using numerical techniques, and use it to perturbatively calculate the dipolar relaxation rate with Fermi's golden rule (2). In Fig. 1 we show how dipolar repulsion (Fig. 1a) modifies the incoming wave function (Fig. 1b) and reduces the integral of the transition matrix element (Fig. 1c).

Without axial excitation, only double spin flips to the final spin state $|j_2\rangle = |7,7\rangle$ and orbital state $m_f = 2$ are allowed. At sufficiently high magnetic field the energy released during the collision can exceed $\hbar\omega_z$, thereby opening up new collisional channels resulting in axial

excitations. Energy conservation requires

$$\frac{\hbar^2 k_f^2}{2\mu} = \frac{\hbar^2 k_i^2}{2\mu} + \Delta j\mu_B g_J B - \Delta n\hbar\omega_z. \qquad (7)$$

The single spin-flip channel ($\Delta j = 1$) requires odd $\Delta n$ due to the odd symmetry of the $\bar{z}$ term in equation (4), whereas double spin flips ($\Delta j = 2$) require even $\Delta n$. Newly opened channels increase the decay rate, as shown in Fig. 3a, b. Furthermore, as previously explained, they also decrease the shielding factor, as visible in the small notch in Fig. 3c.

Remaining in the ground state of the harmonic oscillator is therefore necessary for obtaining extremely low relaxation rates, but that requires working at low enough fields. Unfortunately, the relaxation rates we measure at very low fields deviate from the theoretical values in Fig. 2b, most likely because of imperfect circular polarization of the lattice and compensating beams. The mixture of $\sigma^+$ and $\sigma^-$ light induces Raman couplings between $|m_J = +8\rangle$ and other even $|m_J\rangle$ states, thereby opening additional relaxation channels via spin exchange[21]. With a > 95% circular polarization purity, we find agreement between experimental decay rates and calculated dipolar relaxation rates for fields > 100 mG, where the Raman coupling is suppressed by Zeeman detuning.

## Discussion

We have shown that confinement in thin layers not only reduces the number of available collisional channels, but additionally provides dipolar shielding, thereby strongly suppressing dipolar relaxation between atoms. In principle, arbitrarily low loss rates and infinite shielding factors are possible at very low temperatures. Strong magnetic fields are also predicted to reduce the shielded collision rate to arbitrary low values if strong axial confinement suppresses the opening of collision channels. As we have discussed above, rather straightforward improvements in axial confinement, purity of polarization and temperature should result in rate coefficients in the $10^{-14}$ cm$^3$/s regime.

Our simulations and experiments show that there is already substantial shielding at thermal energies comparable to the barrier height. Lowering the temperature well below the barrier eventually results in exponential suppression[8]. For our experimental parameters, going from 1 μK to 100 nK would increase the suppression by a factor of three.

In this work, we have discussed the interplay between the elastic and inelastic aspects of dipolar interactions. Both scale with the dipolar length, which could be 10,000 times larger for polar molecules. Yet the large total angular momentum $J = 8$ works in favors of dysprosium over molecules, as the elastic part of the dipole-dipole potential scales as $J^4$ in a stretched state, while the relaxation rate scales as $J^3$ for single spin-flips and $J^2$ for double spin-flips.

An important point of comparison is the elastic scattering rate. At 1 Gauss in a trap with a 2 MHz axial frequency, the inelastic 2D cross-section would be 20 nm without shielding. Shielding drops this number to 0.3 nm, while the semi-classical dipolar elastic collisional cross section is $\sigma_{\text{SC}} = 180$ nm[24]. Shielding is necessary to obtain a ratio of good to bad collisions in excess of 100. Shielding is also necessary to study dipolar exchange. Given our density $n_{2D} = 2.9 \times 10^9$ cm$^{-2}$, an estimated spin exchange rate is 200/s, which is comparable to the observed shielded dipolar decay rates.

Dipolar shielding has previously been observed in polar molecules with fermionic statistics[5], for which the shielding is qualitatively different. Since identical fermions already have an isotropic $p$-wave barrier, adding moderate dipolar interactions in a confined geometry will first strengthen this barrier in the radial direction but also weaken it in the axial one. As a result, the inelastic collision rate will first decrease with the dipole moment and then increase[25]. This cannot be seen with

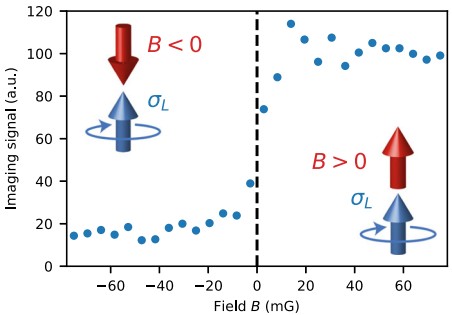

**Fig. 4 | Determining the zero of the magnetic field.** For spin-polarized $m_J = -8$ dysprosium atoms and left-circularly polarized imaging light, the drastic difference in Clebsch-Gordan coefficients for $\sigma_+$ and $\sigma_-$ transitions produces a step-like change in imaging signal as the magnetic field traverses through zero.

bosons. For both particle types, the inelastic collision rate will eventually decrease when entering more deeply into the 2D regime, as we have explored in this work. Our technique would be crucial to study spin mixtures of bulk gases of bosonic dipolar species.

In conclusion, we have demonstrated a way to realize long-lived spin mixtures in dense bosonic lanthanide clouds, opening up new possibilities for quantum simulation experiments in two dimensions. With such technique, dysprosium can be used to study quantum materials with dipolar interactions in regimes different from those currently possible for polar molecules[7] and Rydberg atoms[26]. Stable spin mixtures are important for implementing spin-orbit coupling and artificial gauge potentials via Raman coupling of spin states[27,28]. By suppressing dipolar relaxation, one can take advantage of the ground state orbital angular momentum of lanthanides to avoid the substantial photon scattering rates of the Raman beams for alkali atoms[29].

## Methods
### Sample preparation
The samples are obtained after evaporative cooling in a crossed optical-dipole trap (ODT) which is loaded from the narrow-line magneto-optical trap described in reference[30]. The ODT consists of three 1064 nm laser beams: two beams with 40 $\mu$m beam waists crossed at 8° in the horizontal plane, and a beam with a 64 $\mu$m waist propagating along the (vertical) $z$ direction. We prepare spin-polarized samples of ~ 8 × 10$^4$ $^{162}$Dy atoms in the $|J = 8, m_J = -8\rangle$ state in an optical dipole trap just above the transition temperature. Working with a thermal gas makes it easier to determine dipolar relaxation rate coefficients without accounting for a varying condensate fraction.

The highest spin state $|m_J = +8\rangle$ is populated via adiabatic rapid passage using an RF sweep in a magnetic field of 3.5 G along the $z$ direction. A stack of quasi-2D layers, which we refer to as crêpes due to their extreme aspect ratio, is created using a 1D optical lattice formed by retroreflecting a 741 nm laser beam along the $z$ axis. We use a Ti:Sapph laser focused down to a waist of 50 $\mu$m to the atoms. It can deliver about 300 mW of light after fiber coupling and intensity stabilization. It is typically detuned by 14.25 to 2.25 GHz to the blue side of the narrow 1.8 kHz transition[31], thus providing frequency-controllable tight axial confinement. During the dipolar relaxation experiment, the horizontal beams are switched off, and the vertical 8 W ODT beam serves to compensate for the deconfinement of the blue-detuned lattice. We verified with in-situ images (obtained with detuned imaging light due to the high optical densities) that the blue-detuned lattice is correctly compensated without displacement of the cloud. The lattice and the vertical dipole trap are turned on using exponential ramps with a 50ms time constant to adiabatically load the atoms into the lowest vibrational level of the 2D layers. During the first 40 ms of the lattice ramp, the magnetic field is rapidly reduced to 40mG to minimize the dipolar relaxation losses. The magnetic field is then ramped

up to its final value during the last 10ms of the lattice loading ramp, after which the decay of the sample due to inelastic collisions is measured.

The RMS extension of the cloud along the lattice direction before loading is $\sigma_{ODT} \simeq 4.7\,\mu$m. Given the layer separation of $\lambda/2 \simeq 371$ nm, around $4\sqrt{\pi}\sigma_{ODT}/\lambda = 45$ crêpes are loaded with initially $3 \times 10^4$ atoms and a central density of $n_0 = 2.9 \times 10^9$ cm$^{-2}$. The density distribution in the $i^{th}$ pancake is described by (see Fig. 5)

$$n_i(t,\rho,z) = n_0(t) \exp\left(-z_i^2/(2\sigma_{ODT}^2)\right) \exp\left(-\rho^2/(2\sigma_\perp^2)\right) \quad (8)$$

with $z_i = i\frac{\lambda}{2}$ and $\sigma_\perp = \sqrt{\frac{k_B T_{lattice}}{2\mu\omega_\perp^2}}$, $\sigma_{ODT} = \sqrt{\frac{k_B T_{ODT}}{2\mu\omega_{ODT}^2}}$. The parameters $\omega_{ODT} = 2\pi \cdot 94$ Hz and $T_{ODT} = 150$ nK describe the cloud before the lattice is ramped up whereas $\omega_\perp = 2\pi \cdot 200$ Hz and $T_{lattice} \simeq 1\,\mu$K characterize the conditions after lattice ramp up. The central crêpe contains about 900 atoms. The RMS width of the crêpes is typically $\sigma_z \simeq 10$ nm while the radial one is $\sigma_\perp = 5.7\,\mu$m.

### Zeroing the magnetic field
Achieving control of low magnetic fields is critical for minimizing dipolar suppression by preventing higher outgoing vibrational channels from opening. We have devised a method to zero the magnetic field that relies on the large disparity of Clebsch-Gordan coefficients for dysprosium. When an atom's magnetic moment is aligned along the propagation of a circularly polarized imaging beam, the amount of scattered light strongly differs whether the magnetic dipole moment is oriented parallel or anti-parallel to the propagation of the imaging beam. By using absorption imaging for various external magnetic fields, as shown in Fig. 4, one can observe when the dipole moment has flipped, which determines the zero of the external magnetic field.

More specifically, in a spin-polarized ($m_J = -8$) sample of bosonic dysprosium, the Clebsch-Gordan coefficients for $\sigma_-$, $\pi$ and $\sigma_+$ transitions are 1, 1/9 and 1/153 respectively. We perform absorption imaging of a spin-polarized sample with left-circularly polarized ($\sigma_L$) light along the magnetic field quantization axis $z$. We work with low enough light intensity and imaging time to prevent optical pumping. At large positive magnetic field bias, the atoms see $\sigma_-$ light with a corresponding Clebsch-Gordan coefficient of 1, resulting in a large atom count. At large negative magnetic field bias, the atoms see $\sigma_+$ light with a corresponding Clebsch-Gordan coefficient of 1/153 leading to a low atom count. The lower the transverse magnetic field, the sharper is the transition when the longitudinal field is varied. In this way, the zero settings for all components of the magnetic field are determined.

### Lattice light choice
The need for deep optical lattices requires a tightly focused lattice beam, which causes undesirably strong radial confinement if one uses a red-detuned beam. By choosing a blue-detuned lattice we avoid adiabatic compression of the cloud in the transverse direction and the substantial corresponding increase in temperature when ramping up the optical lattice. The choice of a blue-detuned lattice also exposes the atoms to lower light intensities and reduces the unwanted Raman transitions due to imperfect circular polarization. However, the radial deconfinement created by the lattice needs to be compensated, which we achieve with a red-detuned optical dipole trap that enables independent control of the axial and transverse trap frequencies (see Fig. 5 left).

### Lifetime analysis
The decay of the cloud can be described via equation (1) for the 3D densities

$$\frac{dn_{3D}}{dt} = -\beta_{3D}n_{3D}^2. \quad (9)$$

or by using a 2D equation

$$\frac{dn_{2D}}{dt} = -\beta_{2D} n_{2D}^2. \tag{10}$$

The densities in each pancake are related by

$$n_{3D} = n_{2D} \frac{1}{\sqrt{2\pi}\sigma_z} \exp\left(-z^2/(2\sigma_z^2)\right) \tag{11}$$

with $\sigma_z = \sqrt{\frac{\hbar}{4\mu\omega_z}} = a_z/2 \simeq 10$ nm. Our 2D density of $n_{2D} = 2.9 \times 10^9$ cm$^{-2}$ corresponds to a 3D density of $n_{3D} = 1.1 \times 10^{15}$ cm$^{-3}$. When integrating equation (9) and equating it to (10), we obtain

$$\beta_{3D} = 2\sqrt{\pi}\sigma_z \beta_{2D}. \tag{12}$$

In the main paper, we are using $\beta_{3D}$ to characterize the decay.

We will omit the 2D subscript for the densities in the rest of the manuscript. Equation (10)– in a local-density approximation – needs to be integrated over the cloud volume to relate to the observed quantity

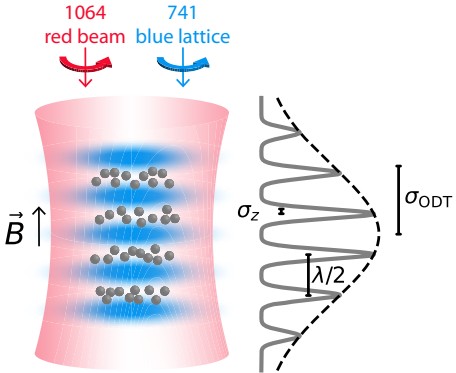

**Fig. 5 | Trap geometry and relevant length scales.** Left: Reproduction of Fig. 2a of the main text. Right: The spatial density of the cloud in the longitudinal direction is characterized by the axial RMS width $\sigma_z = a_z/2$, the lattice spacing $\lambda/2$ and the initial width of the loaded thermal cloud $\sigma_{ODT}$.

$N$, the number of atoms:

$$\frac{dN}{dt} = -\beta_{2D} \int_{\mathcal{S}} n^2 d\tau = -\beta_{2D}\langle n \rangle \equiv -\beta_{2D}\frac{N^2}{V_{eff}}. \tag{13}$$

The effective volume $V_{eff}$ is determined as follows. After integration of equation (8), one gets $N = \sum_i N_i = \sum_i \int n_i \, d\tau = n_0 2\pi\sigma_\perp^2 \sqrt{2\pi}\sigma_{ODT}/(\lambda/2)$ and

$$\frac{dN}{dt} = \sum_i \frac{dN_i}{dt} = -\beta_{2D}\sum_i \int_{\mathcal{S}} n_i^2 \, 2\pi\rho \, d\rho = -\beta_{2D} n_0^2 \pi\sigma_\perp^2 \sum_i e^{-\frac{z_i^2}{2\sigma_{ODT}^2}}$$
$$= -\beta_{2D} n_0^2 \pi\sigma_\perp^2 \frac{\sqrt{\pi}\sigma_{ODT}}{\lambda/2}. \tag{14}$$

Identifying $V_{eff}$ in equation (13) gives

$$V_{eff} = 4\pi\sigma_\perp^2 \frac{2\sqrt{\pi}\sigma_{ODT}}{\lambda/2}. \tag{15}$$

We note that $\langle n \rangle = \frac{N}{2^{3/2} V_{eff}}$, where each $\sqrt{2}$ factor comes from the Gaussian averaging along one axis. To take into account the moderate heating during the experiment, we perform a linear fit of the temperature $T(t) = T_0 + v_T t$ for each measurement of the decay rate, which is used to scale the effective volume $V_{eff}(t)$. The solution of the differential equation (13) that we fit is $N(t) = \frac{N_0}{1 + \frac{\beta_{2D}}{V_{eff}}N_0\frac{\ln(1+v_T t)}{v_T}}$ from which we determine $\beta_{2D}$. A comparison of this fit with one ignoring the temperature increase is shown in Fig. 6. The atom number $N(t)$ is measured as a function of hold time (typically tens of ms) using time of flight imaging.

Here we have assumed that every dipolar relaxation event leads to the loss of both atoms. This is justified since the effective trap depth of a few micro-kelvins is negligible compared to the kinetic energy gained by the spin-flip for magnetic fields larger than a few tens of milligauss. Note that the trap depth is much lower than the axial excitation energy $\hbar\omega_z$. This is due to the compensated blue lattice which provides a very weak trap in the transverse direction compared to the tight axial confinement. The experiment is sufficiently fast (tens of ms) such that photon scattering in the lattice, background collisions and residual evaporation are not important.

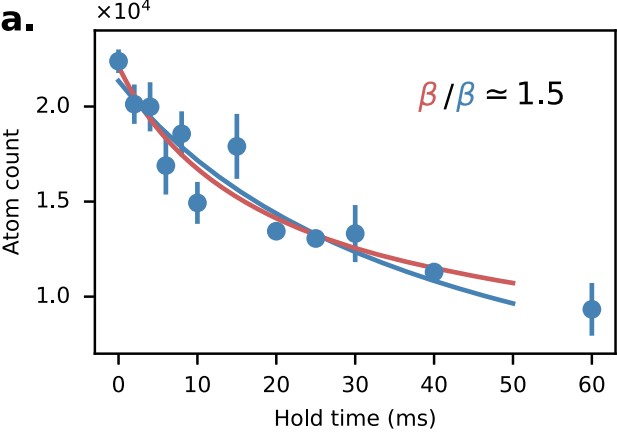

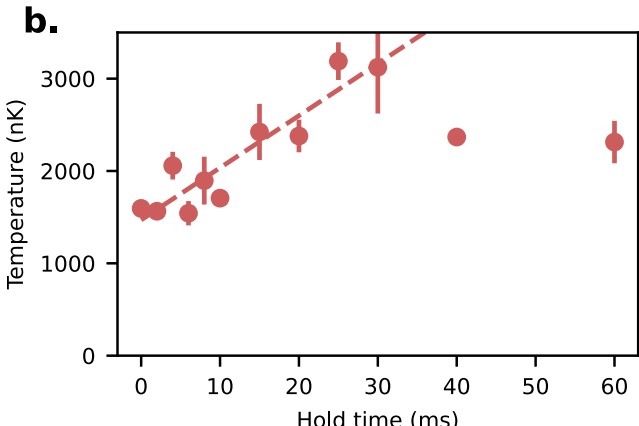

**Fig. 6 | Typical dipolar relaxation decay curve.** The atom number decay is shown in panel **a** and the temperature increase is shown in **b**. Two different decay fit are shown. Either the pure two-body decay (blue) or the one incorporating a linear increase in the temperature, which modifies the density (in red). The difference in the extracted $\beta$ coefficient is here about 50% as incorporating the temperature increase prevents under-fitting the initial fast decay. The loss rate coefficient $\beta$ is systematically 20 to 50% larger with this method. Only the initial part of the temperature increase is used in the fit, as temperature measurements at later times were not reliable. The uncertainty is the statistical standard deviation of the mean of three measurements.

## Error bars

The uncertainties represented by the errorbars in the plots are one standard deviation coming from the statistical error due to the curve fitting, as well as our best estimate in the uncertainties of $\omega_{ODT}$, $T_{ODT}$, $\omega_\perp$, $\omega_z$ and $T_{lattice}$ added in quadrature. $\omega_{ODT}$ is measured in the ODT by observing the oscillation of the cloud after suddenly applying a magnetic force. $\omega_\perp$ is measured by modulating the intensity of the vertical trapping beam and observing the parametric heating resonance. Axial trap frequencies $\omega_z$ (which go up to several hundreds of kHz) are also measured via parametric heating. Due to the bandwidth of the drive electronics, this was done only in shallow lattices and extrapolated to deeper lattices. Temperatures are observed in time of flight.

## Theoretical decay rate

We summarize here the main equations to produce the theoretical predictions in Figs. 1–3. Detailed derivations are given in Supplementary Information Note 2 and ref. 32.

We recall equation (4)

$$\hat{V}_{dd}|j_0\rangle = \frac{\mu_0(Jg_J\mu_B)^2}{4\pi r^3}\left[(1-3\bar{z}^2)|j_0\rangle - \frac{3\bar{z}\bar{r}_+}{J^{1/2}}|j_1\rangle - \frac{3\bar{r}_+^2}{2J}|j_2\rangle\right] \qquad (16)$$

which is used to compute the potential used in equation (6):

$$\left\{\frac{\hbar^2}{2\mu}\left(-\frac{d^2}{d\rho^2} + \frac{m^2-1/4}{\rho^2}\right) + \langle n|V_{dd,0}|n\rangle\right\}\phi = \frac{\hbar^2 k_i^2}{2\mu}\phi. \qquad (17)$$

This is the equation that we solve numerically for both the incoming ($m=0$, $n=0$) and outgoing ($m=2$, $n$ even or $m=1$, $n$ odd) wave functions. The code we developed combines grids of multiple step-sizes to account for the need to appropriately average the potential along $z$, describe the short-range shielding at small $\rho$ and normalize correctly the wave functions at large distances. Given the temperature, magnetic fields, $z$ trapping frequencies and desired precision, the code determines an appropriate grid, and computes the incoming wave function and the harmonic oscillator states on this specific grid. It then distributes those results on multiple cores, computing the outgoing wave function for each of the different decay channels and the respective integral of Fermi's golden rule. This method enables the code to produce the plots presented in this paper on a simple laptop in a reasonable time.

The normalization condition reads: $\int_0^L d\rho\,\phi_{n,m}(\rho)^2 = 1$ for a cylinder of radius $L$. Our model accounts for the modification of both incoming and outgoing wave functions by the dipolar interaction. The free radial wave function solution with momentum $k$ is $\phi_{n,m}^{(free)}(\rho) = \sqrt{\frac{\pi k\rho}{L}}J_m(k\rho)$, which does not depend on $n$.

The 2D loss rate coefficient for the channel $|j_0\rangle \rightarrow |j_f\rangle$, $|0\rangle \rightarrow |n_f\rangle$ reads

$$\beta_{2D}^{j_f,n_f} = \frac{8\mu}{k_i k_f \hbar^3}\left|L\int_{-\infty}^{+\infty}dz\int_0^L d\rho\,\phi_{n_f j_f}(\rho)\chi_{n_f}(z)V_{dd,j_f}(\rho,z)\chi_0(z)\phi_0(\rho)\right|^2 \qquad (18)$$

with $\chi_n$ being the $n^{th}$ harmonic oscillator's state wave function:

$$\chi_n(z) = \frac{1}{\sqrt{2^n n!}}\left(\frac{1}{\pi a_z^2}\right)^{1/4}H_n(z/a_z)e^{-\frac{z^2}{2a_z^2}}. \qquad (19)$$

The total rate is then the sum over all channels:

$$\beta_{2D} = \sum_{j_f,n_f}\beta_{2D}^{j_f,n_f}. \qquad (20)$$

and relates to the 3D rate as $\beta_{3D} = \sqrt{\pi}a_z\beta_{2D}$.

The rate is eventually averaged over the thermal distribution of incoming momenta (see Supplementary Information Note 2) for the Fig. 2b, c, and computed at the mean momentum for all of the other figures.

## Pure-2D limit

In pure-2D the double spin-flip potential is $V_{dd,2}(\rho) \propto 1/\rho^3$. If we ignore the shielding, in the low-temperature limit, we find that the 2D decay rate is

$$\beta_{free}^{pure-2D} = 4\pi^2\frac{1}{J^2}\frac{E_{dd}}{\hbar}a_{dd}^4 k_f^2. \qquad (21)$$

So $\beta_{free}^{pure-2D} \propto B$. If we incorporate shielding we find that

$$\beta_{shielded}^{pure-2D} \propto (1/\log(k_i))^2 \qquad (22)$$

which goes to zero at zero temperature. Under certains assumptions detailed in Supplementary Information Note 4 and noting $x_2$ the first zero of the Bessel function $J_2(x)$, one can find that the decay rate in a certain field range behaves as

$$\beta_{shielded}^{pure-2D} \propto k_f^{1/4}\exp\left(-2\sqrt{\frac{8a_{dd}k_f}{x_2}}\right), \qquad (23)$$

which vanishes at high magnetic fields.

## Discussion of various approximations

**Unitarity limit.** The perturbative results will get modified when the decay rate approaches the unitary limit. However, in our range of parameters, the decay rates are much smaller than the unitary limit. A $\beta_{3D}$ of high $10^{-12}$ cm³/s corresponds to a $\beta_{2D} = \sigma\hbar k/\mu$ in the low $10^{-6}$cm²/s. This gives a $\sigma k \simeq 0.1 \ll 4$ which puts us safely in the non-unitary regime. Note that the total cross section in 2D is $\sigma = \frac{4}{k}\sum_m\sin^2\delta$, and is dominated by the $s$-wave contribution given our magnetic fields.

**Wave function substitution approximation.** The system is perturbed by two parts of the dipolar potential: one which is diagonal in the spin states basis and therefore elastic, the other part is non-diagonal and causes transitions. Usually, the weakest part of the Hamiltonian should be treated perturbatively. It is the case here since $|\hat{V}_{dd}^{inelastic}/\hat{V}_{dd}^{elastic}|^2 \simeq 1/J = 1/8$ as shown in equation (16). This is why we evaluate the decay rate on the shielded wave function. Following previous treatments[2,19,21], we have not checked the importance of higher-order terms in the perturbation theory.

**Effective potential approximation.** To compute the wave functions we assumed an effective potential obtained by averaging $V_{dd}$ in the $n^{th}$ state of the harmonic oscillator. This is the diabatic limit of a coupled channels calculation. There exists a fully adiabatic method to compute the molecular potential of two interacting dipoles in a quasi-2D geometry[9]. It would mix the harmonic oscillator states but we found it would only affect the wave function at short distances, which is important only at high magnetic fields. In our experiment, $k_f a_{dd}$ remains on the order of 1, and restricting the $z$-motion to the pre-existing harmonic oscillator states is acceptable.

**Fermi's golden rule approximation.** The use of Fermi's golden rule with the original density distribution is valid only if the decay rate is smaller than the other time constants of the system. The relaxation rate $\Gamma = \beta_{3D}n \simeq 10^2$ s⁻¹ is indeed smaller than the collision rate which is around $10^3$ s⁻¹, or the trap frequencies of 200 Hz. This assumption is therefore fulfilled. The system will stay in (quasi-) equilibrium when the

loss rates are smaller than the trapping frequencies and smaller than the rate of elastic collisions which provides thermalization.

**Neglecting short-range interactions.** The background $s$-wave scattering length of dysprosium $a = 5.9$ nm[33] can modify the wave functions. A previous paper[2] studied extensively its influence on chromium. However, the decay rate we observed in a large volume 3D trap (red curve in Fig. 2) agrees better with the theory which does not take the scattering length into account. Another dysprosium experiment[19] found a similar result in an even wider range of fields. However, it is possible that the short-range molecular potential plays a role in the 2D results, and could possibly explain why we obtain rates a few times smaller than the theory predicts (see shaded areas in Fig. 2). Indeed, a sizeable contribution to the loss comes from interatomic distances smaller than the van der Waals length $a_{vdW} = 4.3$ nm and the scattering length $a = 5.9$ nm. Since the real wave function rapidly oscillates at short-range, the contribution to the overlap matrix element should vanish. To get a sense of the sensitivity of our model to contact interactions we also used simulated wave functions with a hard-core potential at $a_s = 5.9$ nm, while keeping the dipolar potential elsewhere. We put a node in the incoming and outgoing radial wave functions $\phi$ at this position, and integrated from this distance outward. This produced the lower bound of the shaded areas in Fig. 2. It would be interesting to use a more realistic interaction potential to study the impact of short-range interactions, however this goes beyond the scope of this paper.

## Data availability
The data that support the findings of this study are available from the corresponding author upon request.

## Code availability
The code that supports the findings of this study is available from the corresponding author upon request.

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

## Acknowledgements

We thank Brice Bakkali-Hassani, Hanzhen Lin and Yu-Kun Lu for comments on the manuscript. We acknowledge support from the NSF through the Center for Ultracold Atoms and through Grant No. 1506369, the Vannevar-Bush Faculty Fellowship, and an ARO DURIP grant.

## Author contributions

P.B., M.C, L.D., W.L., A.O.J and W.K. designed and constructed the experimental setup, P.B., M.C., L.D. and J.d.H carried out the experimental work, P.B., M.C., L.D., J.d.H and W.K. developed the theoretical

models and simulations, all authors contributed to the writing of the manuscript.

## Competing interests

The authors declare no competing interests.
