## [Peer Review File · Nature Communications]

Suppressing dipolar relaxation in thin layers of dysprosium atomsEditorial Note: This manuscript has been previously reviewed at another journal that is not operating a transparent peer review scheme. This document only contains reviewer comments and rebuttal letters for versions considered at *Nature Communications*.

REVIEWER COMMENTS

Reviewer #2 (Remarks to the Author):

I thank the author for addressing all my previous comments. I find their responses and changes satisfactory overall. I would like to make one last comment to the authors and suggest that they further clarify the distinction between the suppression mechanism at play in their settings and that of ref. [2]. I believe this would be most beneficial for the audience to understand the novelty of their work. For example, it would be beneficial to explain more directly the principle of dipolar shielding in one sentence in the introduction (instead, the first sentence of the 3rd paragraph could be in part removed), and possibly also in the abstract. The distinction could be highlighted and discussed more extensively than is currently done when discussing Figure 2(b-c). Finally, the second section could also be renamed "Basic Principles *of Dipolar Shielding*". Apart from this comment, I recommend the paper for publication in Nature Communication.

Reviewer #3 (Remarks to the Author):

I am satisfied with the authors' response to my criticism. I recommend publication. Nevertheless, I would like to clarify one point for the sake of rigour.

The authors write

``The common exponential suppression arises from the same shielding mechanism, however the factors differ as they address very different types of losses (short-ranged versus dipolar)."

I disagree with the second part of this statement. The factors differ because the incoming wave function and, therefore, the tunneling amplitude depends on the approximation. The factor $a_{dd}^{(1/2)}$ follows from the WKB treatment of the repulsive $1/\rho^3$ potential and is an artifact of the projection to the oscillator ground state in the z direction. Although the Gaussian assumption is good for calculating the

elastic amplitude, it is not great for calculating the tunneling for large ratios of the dipole length to the oscillator length. In this regime the 3D treatment is essential since the WKB trajectory rather strongly deviates from the plane $z=0$. Please, look more carefully at Buechler et al. (Ref.[4] of the supplement). See, in particular, their Eq.(1) and the corresponding discussion. The exotic scaling $a_{dd}^{2/5}$ comes from the emergent characteristic lengthscale l_{perp} , which is larger than the oscillator length in the limit under discussion. The 2D description is valid for $\rho \gg l_{\text{perp}}$, and at shorter distances the problem is three dimensional.

Reviewer #2 (Remarks to the Author):

I thank the author for addressing all my previous comments. I find their responses and changes satisfactory overall. I would like to make one last comment to the authors and suggest that they further clarify the distinction between the suppression mechanism at play in their settings and that of ref. [2]. I believe this would be most beneficial for the audience to understand the novelty of their work. For example, it would be beneficial to explain more directly the principle of dipolar shielding in one sentence in the introduction (instead, the first sentence of the 3rd paragraph could be in part removed), and possibly also in the abstract. The distinction could be highlighted and discussed more extensively than is currently done when discussing Figure 2(b-c). Finally, the second section could also be renamed "Basic Principles of Dipolar Shielding". Apart from this comment, I recommend the paper for publication in Nature Communication.

We integrated all those suggestions. In particular we emphasized the difference between our mechanism and the one of ref. [2] on pages 2 and 5. The introduction now features an overview of the shielding mechanism, and we renamed the second section.

Reviewer #3 (Remarks to the Author):

I am satisfied with the authors' response to my criticism. I recommend publication. Nevertheless, I would like to clarify one point for the sake of rigour.

The authors write

"The common exponential suppression arises from the same shielding mechanism, however the factors differ as they address very different types of losses (short-ranged versus dipolar)."

I disagree with the second part of this statement. The factors differ because the incoming wave function and, therefore, the tunneling amplitude depends on the approximation. The factor $a_{dd}^{(1/2)}$ follows from the WKB treatment of the repulsive $1/\rho^3$ potential and is an artifact of the projection to the oscillator ground state in the z direction. Although the Gaussian assumption is good for calculating the elastic amplitude, it is not great for calculating the tunneling for large ratios of the dipole length to the oscillator length. In this regime the 3D treatment is essential since the WKB trajectory rather strongly deviates from the plane $z=0$. Please, look more carefully at Buechler et al. (Ref.[4] of the supplement). See, in particular, their Eq.(1) and the corresponding discussion. The exotic scaling $a_{dd}^{(2/5)}$ comes from the emergent characteristic lengthscale l_{\perp} , which is larger than the oscillator length in the limit under discussion. The 2D description is valid for $\rho \gg l_{\perp}$, and at shorter distances the problem is three dimensional.

We thank the reviewer for the correction. It is essential to consider the different geometries of the problems. We now wrote: "The common exponential suppression arises from the same shielding mechanism, however the factors differ as the systems are different: in our case, particles collide at finite momentum in a pure-2D geometry, whereas in the previous works the collision occurs in 3D and at zero temperature. Furthermore, they computed the tunneling rate through the dipolar barrier without assuming any loss mechanism, while we look at the decay rate mediated by the dipolar interaction which extends in our model up to $k_{\perp} \sim k_f$."

REVIEWERS' COMMENTS

Reviewer #3

The reviewer provided confidential remarks to the editor recommending publication of the manuscript.